# A Type II hybrid effectiveness-implementation study of an integrated CHW intervention to address maternal healthcare in rural Nepal

Aparna Tiwari[1]*, Aradhana Thapa[1], Nandini Choudhury[1,2], Rekha Khatri[3], Sabitri Sapkota[2,3], Wan-Ju Wu[1,4,5], Scott Halliday[1,6], David Citrin[1,2,6,7], Ryan Schwarz[1,8,9,10], Duncan Maru[1,2,11,12,13], Hari Jung Rayamazi[14], Rashmi Paudel[14], Laxman Datt Bhatt[14], Ved Bhandari[14], Nutan Marasini[14], Sonu Khadka[15], Bhawana Bogati[14], Sita Saud[16], Yashoda Kumari Bhat Kshetri[17], Aasha Bhatta[18], Kshitiz Rana Magar[19], Ramesh Shrestha[19], Ranjana Kafle[20], Roshan Poudel[14], Samiksha Gautam[14], Indira Basnett[3], Goma Niroula Shrestha[21], Isha Nirola[22], Samrachana Adhikari[23], Poshan Thapa[24], Lal Kunwar[25], Sheela Maru[1,2,11,26]

1 Possible, New York, NY, United States of America, 2 Icahn School of Medicine at Mount Sinai, Arnhold Institute for Global Health, New York, NY, United States of America, 3 Possible, Kathmandu, Nepal, 4 Department of Obstetrics and Gynecology, Boston Medical Center, Boston, MA, United States of America, 5 Department of Obstetrics and Gynecology, Boston University School of Medicine, Boston, MA, United States of America, 6 Department of Global Health, University of Washington, Seattle, WA, United States of America, 7 Department of Anthropology, University of Washington, Seattle, WA, United States of America, 8 Department of Medicine, Division of Global Health Equity, Brigham and Women's Hospital, Boston, MA, United States of America, 9 Department of Medicine, Harvard Medical School, Boston, MA, United States of America, 10 Department of Medicine, Division of General Internal Medicine, Massachusetts General Hospital, Boston, MA, United States of America, 11 Department of Health Systems Design and Global Health, Icahn School of Medicine at Mount Sinai, New York, United States of America, 12 Department of Internal Medicine, Icahn School of Medicine at Mount Sinai, New York, NY, United States of America, 13 Department of Pediatrics, Icahn School of Medicine at Mount Sinai, New York, NY, United States of America, 14 Nyaya Health Nepal, Kathmandu, Nepal, 15 Gandaki Medical College Teaching Hospital and Research Center, Pokhara, Nepal, 16 Civil Service Hospital, Kathmandu, Nepal, 17 COVID Hospital, Shikhar Municipality, Doti, Nepal, 18 Amargadhi Municipality, Dadeldhura, Nepal, 19 Department of Public Health and Community Program, Dhulikhel Hospital, Kathmandu University Hospital, Dhulikhel, Nepal, 20 Nick Simons Institute, Lalitpur, Nepal, 21 Department of Health Services, Nursing and Social Security Division (NSSD), Ministry of Health and Population, Nepal, 22 Harvard T.H. Chan School of Public Health, Boston, MA, United States of America, 23 Department of Population Health, NYU School of Medicine, New York, NY, United States of America, 24 University of New South Wales, School of Population Health, Sydney, Australia, 25 Medic, Lalitpur, Nepal, 26 Department of Obstetrics, Gynecology and Reproductive Science, Icahn School of Medicine at Mount Sinai, New York, NY, United States of America

* aparna.tiwari@possiblehealth.org

**Data Availability Statement:** The datasets generated and/or analyzed during the current study

## Abstract

Skilled care during pregnancy, childbirth, and postpartum is essential to prevent adverse maternal health outcomes, yet utilization of care remains low in many resource-limited countries, including Nepal. Community health workers (CHWs) can mitigate health system challenges and geographical barriers to achieving universal health coverage. Gaps remain, however, in understanding whether evidence-based interventions delivered by CHWs, closely aligned with WHO recommendations, are effective in Nepal's context. We conducted a type II hybrid effectiveness-implementation, mixed-methods study in two rural districts in Nepal to evaluate the effectiveness and the implementation of an evidence-based integrated maternal and child health intervention delivered by CHWs, using a mobile

is publicly available in the OSF Repository: https://osf.io/quhzv/.

**Funding:** This work was supported by Grand Challenges Canada [GCC Grant Number 1808-17775 and TTS-2009-35989], United States Agency for International Development via a Partnerships for Enhanced Engagement in Research award [sponsor grant number AID-OAA-A11-00012, National Academy of Science subaward letter 2000007780] and the Office of the Director, National Institutes of Health under an Early Independence Award [DP5OD019894] to DM (the Eunice Kennedy Shriver National Institute Of Child Health & Human Development (NICHD) and the National Institute Of Dental & Craniofacial Research (NIDCR)). The funders had no role in study design, data collection and analysis, decision to publish, or preparation of the manuscript. Any findings, conclusions, or recommendations expressed in this article are those of the authors alone and do not necessarily reflect the views of the Grand Challenges Canada, United States Agency for International Development, National Institutes of Health or the National Academy of Science.

**Competing interests:** I have read the journal's policy and the authors of this manuscript have the following competing interests: A. Tiwari and A. Thapa are employed by a US-based non-profit (Possible) and based in Nepal. S. Sapkota and R. Khatri are employed by a Nepal-based non-governmental organization, Possible that operates with support from US-based Possible. VB, BB, HJR, R. Paudel, SG, NM, R. Poudel and LDB are employed by a Nepal-based non-governmental organization (Nyaya Health Nepal) that delivers free healthcare in rural Nepal using funds from the Government of Nepal and other public, philanthropic, and private foundation sources. NC, DM, SM are employed by, and SM, DC, DM, and S. Sapkota are faculty members at a private medical school (Icahn School of Medicine at Mount Sinai). DM is a member on US-based Possible's Board of Directors, for which he receives no compensation. IB is a board chair of Nepal-based Possible. WW is a faculty member at a private university (Boston University School of Medicine). DC is a faculty member and SH is a graduate student at a public university (University of Washington). R. Schwarz is employed at an academic medical center (Brigham and Women's Hospital) that receives public sector research funding, as well as revenue through private sector fee-for-service medical transactions and private foundation grants. R. Schwarz is a faculty member at a private medical school (Harvard Medical School) and employed at

application. The intervention was implemented stepwise over four years (2014–2018), with 65 CHWs enrolling 30,785 families. We performed a mixed-effects Poisson regression to assess institutional birth rate (IBR) pre-and post-intervention. We used the Reach, Effectiveness, Adoption, Implementation, and Maintenance framework to evaluate the implementation during and after the study completion. There was an average 30% increase in IBR post-intervention, adjusting for confounding variables (p<0.0001). Study enrollment showed 35% of families identified as *dalit*, *janjati*, or other castes. About 78–89% of postpartum women received at least one CHW-counseled home visit within 60 days of childbirth. Ten (53% of planned) municipalities adopted the intervention during the study period. Implementation fidelity, measured by median counseled home visits, improved with intervention time. The intervention was institutionalized beyond the study period and expanded to four additional hubs, albeit with adjustments in management and supervision. Mechanisms of intervention impact include increased knowledge, timely referrals, and longitudinal CHW interaction. Full-time, supervised, and trained CHWs delivering evidence-based integrated care appears to be effective in improving maternal healthcare in rural Nepal. This study contributes to the growing body of evidence on the role of community health workers in achieving universal health coverage.

## Introduction

Maternal mortality remains high in most low- and middle-income countries (LMICs) despite positive trends over time. About 810 maternal deaths occur every day globally due to preventable pregnancy- or childbirth-related conditions, with LMICs accounting for 94% of all maternal deaths [1]. Skilled care during childbirth and follow-up in the immediate postnatal period are important strategies to prevent maternal and neonatal morbidity and mortality [2]. Utilization of institutional delivery services and critical early postnatal care, however, remains low in many LMICs [2–4]. While encouraging institutional birth is essential to address the direct causes of maternal mortality, this approach alone is not sufficient for improving maternal health outcomes [5]. Interventions integrated across the continuum of care are most effective in preventing adverse maternal health outcomes [2,6]. It is also essential to address the complex contextual challenges, such as financial incentives, effects of decentralization, and health systems' challenges, including workforce shortages in resource-limited settings [5,7]. Community health workers (CHWs) are increasingly recognized as a promising cadre to expand access to health services and achieve universal health coverage (UHC) in resource-limited settings [8,9].

Similar to other resource-limited countries, Nepal grapples with health system challenges in achieving UHC [10]. Despite reducing the maternal mortality ratio from 850 deaths per 100,000 live births in 1990 to 239 in 2015, Nepal still lags in achieving the Sustainable Development Goals (SDGs) target of 70 deaths per 100,000 live births by 2030 [11,12]. Nationally, the utilization of essential maternal healthcare services remains low, with the institutional birth and four facility antenatal care completion rates at 51% and 69%, respectively [11]. Regional disparities in health care access and outcomes also exist between rural and urban areas [11]. Nepal's hilly terrains also pose challenges in accessing essential maternal healthcare services in remote settings [13]. To mitigate the physical barriers to access, Nepal had deployed various community-based cadres in the past, including the existing female community health volunteers (FCHVs) [14,15]. While these lay FCHVs played pivotal roles in improving health outcomes, they have faced several challenges with training, supervision, motivation, and

an academic medical center (Massachusetts General Hospital) that receives public sector research funding, as well as revenue through private sector fee-for-service medical transactions and private foundation grants. SA is a faculty member at a private medical school (NYU School of Medicine). SK is a nursing student at Gandaki Medical College Teaching Hospital and Research Center. IN is a graduate student at a private university (Harvard T.H. Chan School of Public Health). LK is employed by a non-profit (Medic). PT is a graduate student at a public university (University of New South Wales). S. Saud is employed by a government hospital of Nepal (Civil Service Hospital). YKBK is employed by a government hospital of Nepal (COVID Hospital in Shikhar municipality). AB is employed by the local government of Nepal (Amargadhi municipality). R. Shrestha and KRM are employed by a non-profit hospital (Dhulikhel Hospital, Kathmandu University Hospital). R. Kafle is employed by a Nepal-based non-profit (Nick Simons Institute). GNS is a director of the Nursing and Social Security Division under the Government of Nepal Ministry of Health and Population. All authors declare that we have no competing financial interests.

performance [16]. Despite the continuous efforts of the Government of Nepal (GoN) towards improving this community-based cadre, Nepal will fall short of meeting SDG targets by 2030 [12,14]. Nepal Public Health Service Act 2018 has emphasized fulfilling processes and standards while providing community-based health services prescribed by local governments. As a result of a growing recognition of the need to improve and make additional investments in CHW systems, the GoN has attempted to leverage another cadre, auxiliary nurse midwives, for community-based care in one Far-Western district in Nepal [17]. However, the program had not expanded beyond the pilot area and its impact is unclear.

The World Health Organization (WHO) recommends full-time employment, remuneration, minimum education requirements, appropriate training, and supportive supervision of CHWs for effective delivery of services [18]. There is a gap, however, in understanding whether delivering an evidence-based intervention by a CHW cadre with these characteristics is feasible and effective for maternal healthcare improvements in Nepal's context. Therefore, the Family Welfare Division under the GoN Ministry of Health and Population collaborated with *Nyaya Health Nepal* (NHN), a Nepali non-governmental organization, and *Possible*, a US-based non-profit, to implement and assess the effectiveness of a CHW program closely aligned with WHO guidelines in two rural districts (Achham and Dolakha) to reach an estimated 300,000 catchment area population. CHWs were local women with a minimum of tenth grade education, full-time, salaried, trained, and supervised, in line with the WHO recommendations. They delivered an evidence-based integrated reproductive, maternal, newborn, and child health (RMNCH) intervention. CHWs utilized a mobile platform, CommCare, for home-based counseling and assessments, decision support, and simultaneous data collection [19,20].

The CHW pilot study was a type II hybrid effectiveness-implementation, mixed-methods study that sought to evaluate the implementation and the effectiveness of the evidence-based integrated RMNCH intervention delivered by CHWs in rural Nepal. We evaluated the IBR as the primary effectiveness outcome. To guide our evaluation of the implementation both during and after the completion of the study, we used the Reach, Effectiveness, Adoption, Implementation, and Maintenance (RE-AIM) framework [20,21]. In this paper, we present our findings of the integrated RMNCH intervention delivered by CHWs on IBR and antenatal and postnatal care specific RE-AIM outcomes.

## Methods

### Study site

This study was conducted in Achham and Dolakha districts in Nepal. Located in the Far Western Province, Achham is one of Nepal's most impoverished districts. As a result of its hilly terrain, remoteness, and ten-years-long armed conflicts, Achham has some of the worst health outcomes in the country [22]. Dolakha district, located in the Bagmati Province, while closer to the capital city, has hilly terrain, rural geography, and poor road infrastructure that pose physical barriers to healthcare access. As the epicenter of the second major earthquake in Nepal in 2015, Dolakha was one of the hardest-hit districts, and much of the healthcare infrastructure was destroyed [23].

Primary level facilities, locally known as health posts or birthing centers staffed with health assistants and auxiliary nurse midwives for skilled birth services [24], are accessible to all study sites. Additionally, some of the study sites are close to a Primary Health Care Center and district-level hospital with physician staff, cesarean section capacity, and blood transfusion services [24].

## Study population

The target study population consisted of married women of reproductive age (15–49 years) in Achham and Dolakha. We used a census methodology to recruit all eligible participants from our study catchment areas [25]. Women who provided consent to enroll and receive antenatal and postnatal care and to use their data in the study were included in data analysis. Women who were outside the age range or who did not agree to receive services were excluded. For qualitative data collection, the study population consisted of reproductive-aged women enrolled in the CHW program, CHWs, community health nurses (CHNs), and program staff.

## Intervention

CHWs in our study were full-time paid employees, had a minimum tenth-grade education, and received structured supervision, characteristics which differed from the government FCHVs [19,20,26]. Similar to FCHVs, they were all women residing in the communities they served. CHNs, who had a minimum education of a certificate level in nursing, directly supervised the CHWs during home visits and regular office meetings and provided data review and training (1 CHN: 5 CHWs). Community health program associates (CHPAs) were CHNs' direct supervisors (1 CHPA: 2–3 CHNs) and supported program planning and general administrative tasks [14].

The integrated intervention that CHWs delivered in the communities consisted of five different components: 1) continuous monitoring of all reproductive-aged women for early pregnancy detection; 2) home-based antenatal care (ANC) and postnatal care (PNC); 3) early childhood care of under-two children using the GoN's Community-Based Integrated Management of Newborn and Childhood Illness protocol (CB-IMNCI); 4) Group antenatal care (Group ANC); and 5) person-centered contraceptive counseling using the Balanced Counseling Strategy. The details of the integrated intervention are in the published study protocol. This manuscript focuses on the ANC and PNC components and their evaluation. We will present additional components and findings in a forthcoming manuscript.

In summary, CHWs visited all married women of reproductive age every three months and actively screened for pregnancy using an algorithm built into the mobile tool and urine pregnancy tests. Pregnant women who desired to continue their pregnancies were provided ANC; otherwise, they were referred for safe abortion counseling. CHWs made monthly home visits for ANC and provided gestational month-specific counseling on pregnancy care, nutrition, birth preparedness, postpartum contraception, and newborn care. CHWs also screened for danger signs in pregnancies through a series of prompts in the mobile tool and made referrals as appropriate. In addition to home visits, pregnant women received four Group ANC sessions at the local health post or birthing center, co-facilitated by CHWs and government nurse-midwives. Group ANC sessions included discussions on receiving the government-recommended four ANC visits and pregnancy topics with peers of similar gestational age. CHNs also performed diagnostic lab tests and ultrasound in Group ANC and made referrals for complications. For PNC, CHWs visited postpartum women every month after birth, provided contraceptive counseling, screened for complications, and made appropriate referrals (Fig 1). CHWs also provided early childhood care to newborns using CB-IMNCI along with PNC every month.

CHWs used CommCare, an open-source mobile platform, for counseling all the intervention components, including pregnancy screening, ANC, PNC, and early childhood care. Besides counseling, CommCare also aided CHWs in simultaneous data collection and decision support for referrals. Digital engineers and a monitoring and evaluation team initially trained CHWs for a week on CommCare use for data collection and provided ongoing technical

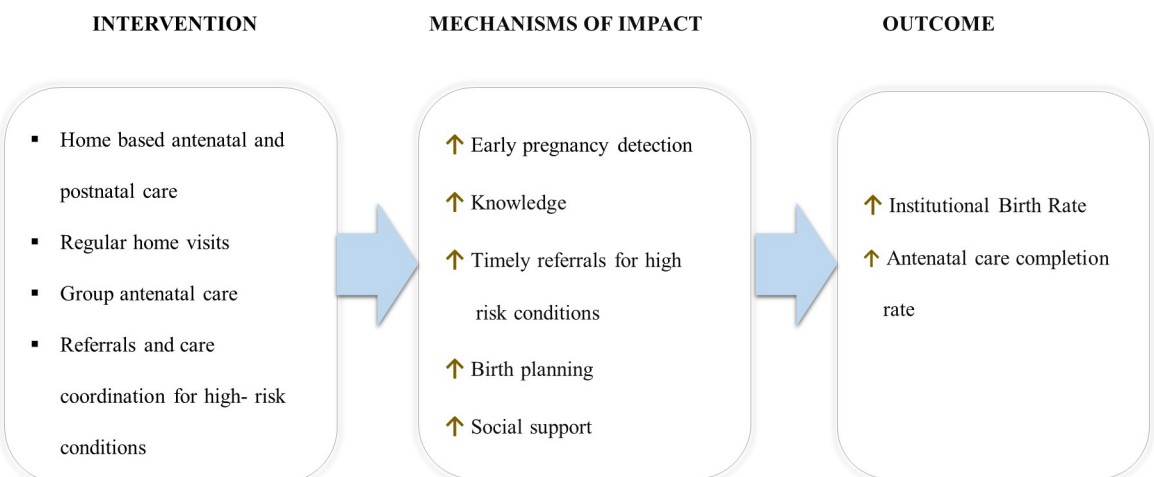

INTERVENTION | MECHANISMS OF IMPACT | OUTCOME

**Fig 1. Conceptual model for antenatal and postnatal components of the integrated intervention.**

support as needed. CHNs and CHPAs provided a month-long pre-service programmatic training and regular refresher training.

## Study design and implementation approach

This study was a type II hybrid effectiveness-implementation study, where we sought to evaluate both implementation and effectiveness outcomes using a mixed-methods approach [20]. It was a non-randomized, pragmatic trial conducted within the constraints of a resource-limited setting and the evolving political context in Nepal. The primary effectiveness outcome of this study was the IBR. We assessed the ward-level IBR pre- and post-intervention to evaluate the effectiveness. A ward in Nepal is a geopolitical unit, the smallest unit of local government. Of note, our analysis was at the ward-level, or population level, since the study did not involve following the same individual women pre- and post- intervention. In addition, our intervention was implemented at the ward-level, with a CHW assigned to each ward. We used the RE-AIM framework to guide the quantitative and qualitative evaluation of the implementation. The RE-AIM framework is commonly used in implementation research to translate scientific knowledge into public health practice by assessing and reporting on the intervention context, generalizability, implementation conditions and process outcomes [21,27]. We also used qualitative methods to understand mechanisms of impact and to triangulate quantitative findings.

The integrated intervention was implemented in a stepped fashion in Achham and Dolakha districts, across seven different hubs, one hub at a time. Hubs defined in our study were clusters of about 2–14 wards. Of note, the hubs in our study do not exactly correspond to the current municipal boundaries since this study began before the institutionalization of the federal system and municipal restructuring that took place in March 2017. It was an intentional design to have the local governments decide on the timing of the program implementation to improve government ownership and scalability in the long term. As such, randomization and simultaneous control groups were not feasible.

Pre-intervention or baseline data collection took place at the time of enrollment of households, families, and individuals into the program. Enrollment for the study, inclusive of all steps, began in November 2014 and continued through September 2018. The enrollment period lasted for approximately 3–4 months in each hub. The intervals *between* enrollment initiation from one hub to the next ranged from 1 to 16 months because enrollment initiation

depended on local government readiness to begin the study and intervention. There was also an interval between enrollment and the rollout of the intervention components *within* each hub, ranging from 1 to 10 months, to allow time for training and implementation preparation.

## Data collection

During enrollment, CHWs visited each house in their assigned wards, placed number plates with a unique identification number after obtaining consent, and enrolled all families, married women of reproductive age, and children under two years. They recorded the enrollment information on mobile platforms. CHWs also collected family demographics and for, married reproductive-aged women, a birth history of the preceding two years. Birth history data were utilized for pre-intervention or baseline. Of note, CHWs used the mobile platform, SurveyCTO in Hub1 and CommCare in all subsequent hubs for baseline data collection.

After enrollment, the integrated intervention was implemented in phases in each hub. First, we implemented continuous monitoring of reproductive-aged women, home-based ANC and PNC, and contraceptive counseling; Group ANC was implemented later (except for in Hub 1, where there had been a prior study of Group ANC alone) [28]. Post-intervention data were from programmatic data collected by CHWs using CommCare. Follow-up data included yearly data from the intervention rollout date. After quantitative data collection concluded in March 2020, the amount of post-intervention data varied by hubs from 1 to 4 years, owing to the stepped implementation.

Between June 2019—October 2020, we conducted a total of 17 semi-structured individual interviews: reproductive-aged women intervention recipients (n = 5), CHNs (n = 3), CHPAs (n = 3), program managers (n = 4), a program director (n = 1), and a monitoring and evaluation associate (n = 1). Program staff were purposively selected based on their programmatic duties and work duration. We also conducted three focus group discussions (FGDs) with CHWs working in 3 geographic areas. Each FGD contained 10–11 CHWs. The geographic areas were purposively selected based on duration of program implementation in that area, and all CHWs working in that area were invited to the FGD. A qualitative researcher, who was not part of the care delivery team, conducted interviews and FGDs in Nepali. They were audio-recorded with participants' permission.

Qualitative data collection took place throughout the study period at different times to understand the implementation and effectiveness within the RE-AIM framework. For reach, we sought to understand barriers to coverage and enrollment of participants, from CHWs and program staff. For primary and secondary effectiveness, we sought to understand the perception of changes in the utilization of care after implementation of the intervention, from women receiving the intervention, CHWs, program staff, and local stakeholders. For adoption, we sought to understand the program adoption and ownership, enablers, and challenges of timely adoption of the program, from local stakeholders, program managers, and CHWs. For implementation, we sought to understand barriers and facilitators to implementation, from local stakeholders, CHWs, program staff, and women receiving the intervention.

## Data analysis

The primary effectiveness indicator of this study was IBR, which we defined as the proportion of all births that occurred in any health facility, public or private. We considered births 'non-institutional' if they took place at home or on the road. We assessed pre-intervention IBR and repeated post-intervention IBR yearly throughout the study period at the ward-level for each hub. We summarized IBR and potential confounding variables using descriptive statistics and graphical summaries. We used bar graphs and box plots to visualize the

potential association between IBR and our covariables (*intervention*, *district*, *intervention years*, *hub*, *ward*, *calendar year*).

We looked at IBR outcomes of women clustered within each ward as opposed to individual IBR. Since our data was count data, we used a mixed-effects Poisson regression model to assess the change in IBR post-intervention compared to pre-intervention. To determine the effects of the intervention on IBR, we included a binary variable *intervention* that was categorized into pre-intervention (for pre-intervention or year 0) and post-intervention (for data collected during intervention years 1, 2, 3, and 4). Assuming the intervention's effects on the outcome were consistent across the catchment areas, we included the *intervention* as a fixed-effect variable in the model. We adjusted for a *district* variable in the model since the two districts are located in different geographic regions with significant differences in social and economic conditions, healthcare access, and pre-intervention IBR. Additionally, we adjusted for a discrete *time period* variable in the model because the number of years of intervention implementation might also influence the outcome, as shown by the IBR and intervention periods boxplots (Fig 3). Of note, the intervention time periods (years 0, 1, 2, 3, 4) of the hubs in our study did not correspond to the same calendar years because of the stepped implementation design. Therefore, we also included a variable representing the approximate calendar year to account for the temporal effects on IBR due to societal development over time. We considered *hub* as a random effect in the model to account for the clustering of wards within a hub.

To select the variables in our final model that best explained the association between the intervention and IBR, we included covariables sequentially starting with our primary exposure variable, *intervention*. We then sequentially added other covariables (*hub*, *district*, *time period*, *and calendar year)*. The model with and without the *calendar year* variable yielded similar parameter estimates with a minimal difference in Akaike Information Criterion values. We used the likelihood ratio test to compare the log-likelihood of the two models. The non-significant statistic indicated the model without the *calendar year* fit our data significantly better. Therefore, we used the parsimonious model as the final model to assess the association between the intervention and IBR after adjusting for confounding variables.

We used the RE-AIM framework to assess the implementation of the intervention using both quantitative metrics and qualitative data. To evaluate reach, we looked at the proportion of families enrolled in the program who were of *dalit*, *janjati*, or other castes, traditionally considered to be at the bottom of the social stratification and had relatively lower socioeconomic status than their 'upper caste' counterparts. To assess the reach of the early postnatal intervention, we evaluated the percentage of one-year postpartum women who received at least one counseled postnatal home visit within a week and 60 days of birth. To assess secondary effectiveness beyond IBR, we evaluated the proportion of women who self-reported four or more ANC visits in health facilities during their pregnancy (the government-recommended number). We assessed the four ANC completion rates pre- and post-intervention at the ward level in each hub. We then performed a Wilcoxon signed-rank test on non-missing ANC completion data to assess whether the change in rates pre- and post-intervention was significant. For program adoption, we measured the percentage of municipalities that adopted the program. For implementation, we examined the median number of home-based CHW ANC visits per full-term pregnancy. We also looked at the number of high-risk pregnancy and postpartum referrals CHWs made. For maintenance, we evaluated the sustainability by assessing the cost of the intervention using data from the first two hubs [14]. We also looked at whether the program was institutionalized beyond the study period and whether there were any adjustments to the pilot program. We performed the Poisson regression using R—3.6.1 and other quantitative analyses using SAS University Edition version 9.4.

For the qualitative analysis, the audio-recorded interviews and FGDs were transcribed and translated into English. The qualitative researcher checked the transcripts to ensure quality and consistency. The transcripts were uploaded into Dedoose (*SocioCultural Research Consultants, LLC 2016*) and coded in English initially using an inductive approach. We extracted codes and data specific to ANC and PNC and then re-coded the data with reference to the RE-AIM framework and our original research questions. Then we categorized the codes into themes and subthemes.

## Ethics approval

The Nepal Health Research Council [(133/2014) and (461/2016)], the Brigham and Women's Hospital [(2017P000709/PHS) and (2015P000058/BWH)], and the Mount Sinai School of Medicine institutional boards (MSSM IRB-18-01091) institutional review boards approved the study for human subjects' research. Boston Medical Center (H-38196) exempted the study.

At enrollment, CHWs read a script in CommCare and obtained verbal informed consent to enroll households and individuals into the CHW program, provide care, and use their data for research. Consent was documented in the mobile application. The qualitative researcher also read a structured script and obtained verbal informed consent from all qualitative interviews and FGD participants.

## Results

### Study enrollment

During enrollment, a total of 31,178 families were identified across all hubs. Of these, 393 (1%) were unavailable at home or declined consent for enrollment into the program. Of 30,785 enrolled families, 35% identified as *dalit*, *janjati*, or other castes. The pre-intervention median monthly household expenses across all hubs ranged from 4000 to 12,000 Nepali rupees (approximately US $34 to $102). The pre-intervention IBR across all hubs ranged from 51% to 78% (Table 1).

We reported the number of unique birth history forms filled at enrollment to approximate the number of enrolled women because baseline individual-level data could not be processed

**Table 1. Socio-demographics of study catchment areas.**

| Characteristics | Total | Hub 1 | Hub 2 | Hub 3 | Hub 4 | Hub 5 | Hub 6 | Hub 7 |
|---|---|---|---|---|---|---|---|---|
| Number of wards, n | 50 | 14 | 8 | 8 | 5 | 2 | 7 | 6 |
| Enrollment period (Month/Year) | -- | Nov 14 –Feb 15 | Jun 16 –Sep 16 | Jul 17 –Nov 17 | Sep 17 –Jan 18 | Jan 18 –Apr 18 | Apr18 –Jul 18 | May 18 –Sep 18 |
| Families enrolled | 30,785 | 8075 | 3550 | 5834 | 3323 | 1013 | 4264 | 4726 |
| *Dalit/janjati/*other families enrolled (n, %) | 10,734 (35%) | 2279 (28%) | 958 (27%) | 2703 (46%) | 947 (28%) | 318 (31%) | 935 (22%) | 2594 (55%) |
| Monthly household expense in NPR Median [Q1, Q3] | -- | 6000 [4000, 10000] | 5000 [3000, 7000] | 10000 [5000, 15000] | 5000 [3000, 7000] | 4000 [2000, 5000] | 6000 [5000, 9000] | 12000 [9000, 20000] |
| Birth centers (with Skilled Birth Attendant) | 32 | 9 | 6 | 4 | 5 | 1 | 5 | 2 |
| CHWs, n | 65 | 19 | 9 | 9 | 7 | 2 | 12 | 7 |
| Baseline institutional birth rate | 70% | 76% | 53% | 78% | 51% | 57% | 74% | 68% |

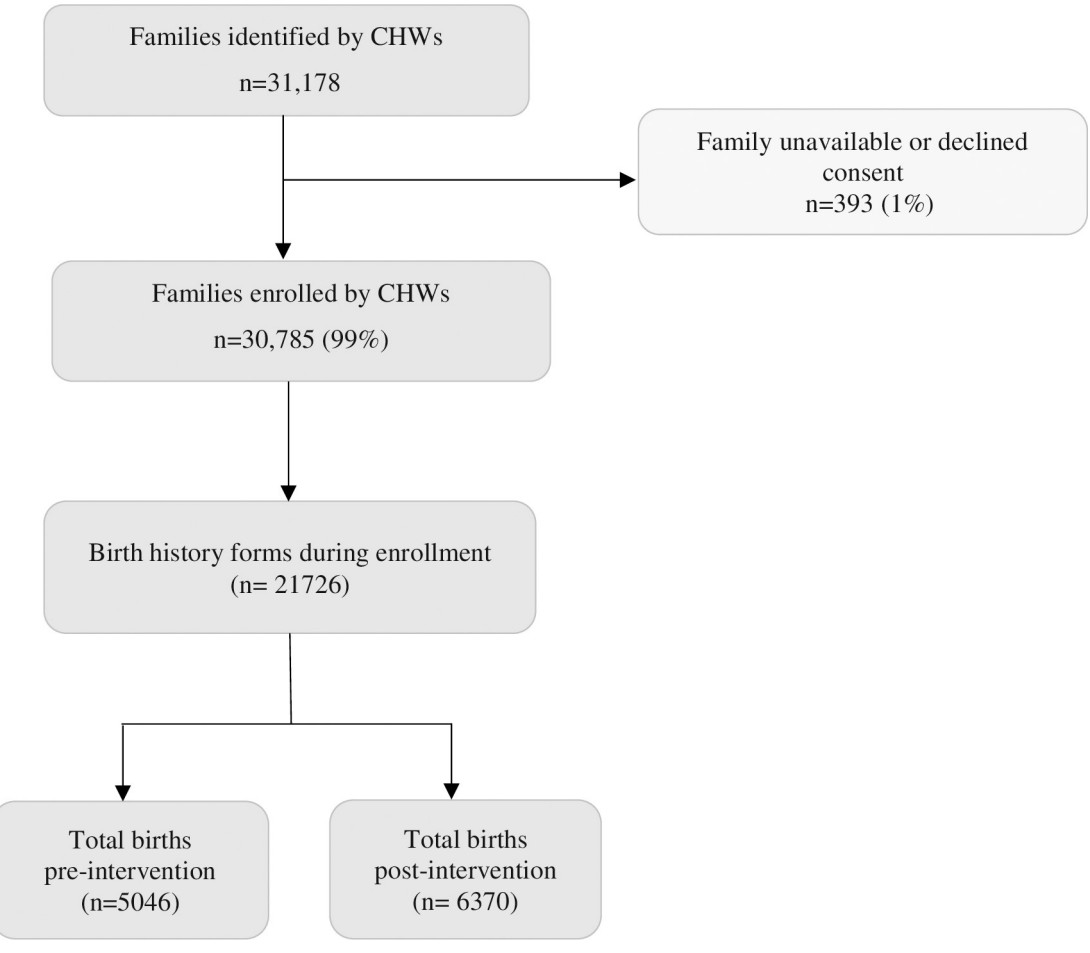

**Fig 2. Study enrollment.**

for three hubs. CHWs filled out birth history forms at enrollment for all enrolled women unless they declined consent. We compared both the total number of women enrolled and birth history forms at enrollment for hubs where both data were available and did not find significant differences (*Hub 4*: women enrolled = 2902 birth history forms = 2894; *Hub 5*: women enrolled = 759, birth history forms = 755; *Hub 6*: women enrolled = 4032 birth history forms = 4021). There were a total of 5046 and 6370 births across all hubs in pre- and post-intervention periods, respectively (Fig 2).

The box plots of unadjusted IBRs by intervention time periods (years 0, 1, 2, 3, 4) showed a potential association between the outcome and the intervention years. There was an increase in IBR one-year post intervention, and the effects seemed to increase over time, although at a much lower rate in the later years (Fig 3). Due to this observed association, we included the intervention years in the regression model.

## Primary effectiveness outcome (institutional birth rate)

Adjusting for districts, intervention years, and random effects of hubs, we used a mixed-effects Poisson regression model to assess IBR pre- and post-intervention. The regression analysis yielded the rate ratio estimate of 1.3 (CI: 1.2, 1.4), a statistically significant association between

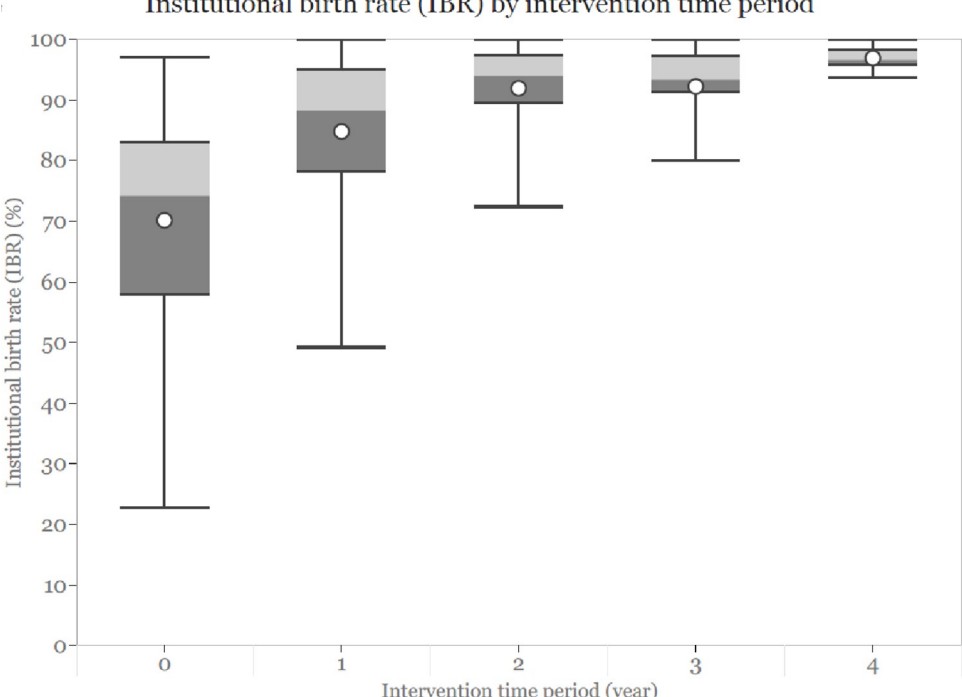

**Fig 3. Unadjusted institutional birth rates by intervention time period.**

IBR and the intervention. There was, on average, a 30% increase in IBR (p<0.0001) post-intervention as compared to pre-intervention, adjusting for other variables (Table 2).

Our qualitative data supported the positive changes in health outcomes with intervention implementation. CHWs perceived the intervention contributed to the increase in institutional deliveries. One CHW stated: *"Earlier, there used to be home delivery, but because of our efforts, most deliveries happen in health institutions. People say that many mothers and children died earlier, but this has improved because of our efforts, and hearing this really encourages us"* (CHW FGD #3). The change in outcomes was also noted by women receiving the intervention: *"[Women] used to give birth in their houses before. They did not go to the hospital, but they go to the hospital nowadays"* (Reproductive-aged women #2).

## Implementation outcomes (RE-AIM framework)

**Reach.** The integrated intervention reached a total catchment area population of approximately 236,000 across both Achham and Dolakha districts during the study period. The program did not expand to the targeted 300,000 catchment area population due to changes in local government priorities. According to the 2014 census, approximately 54% of the

**Table 2. Generalized linear mixed model fit by maximum likelihood (laplace approximation) fixed effects (raw estimates).**

| Parameter | Estimate | Std. error | Z value | Pr. > z |
|---|---|---|---|---|
| Intercept | -0.44770 | 0.03901 | -11.477 | < 2e-16 |
| Intervention (Ref = pre-intervention) | 0.23357 | 0.03424 | 6.821 | 9.06e-12 |
| District (Ref = Dolakha) | 0.07078 | 0.07000 | 1.011 | 0.31195 |
| Time period (Ref = year 0) | 0.04490 | 0.01484 | 3.025 | 0.00248 |

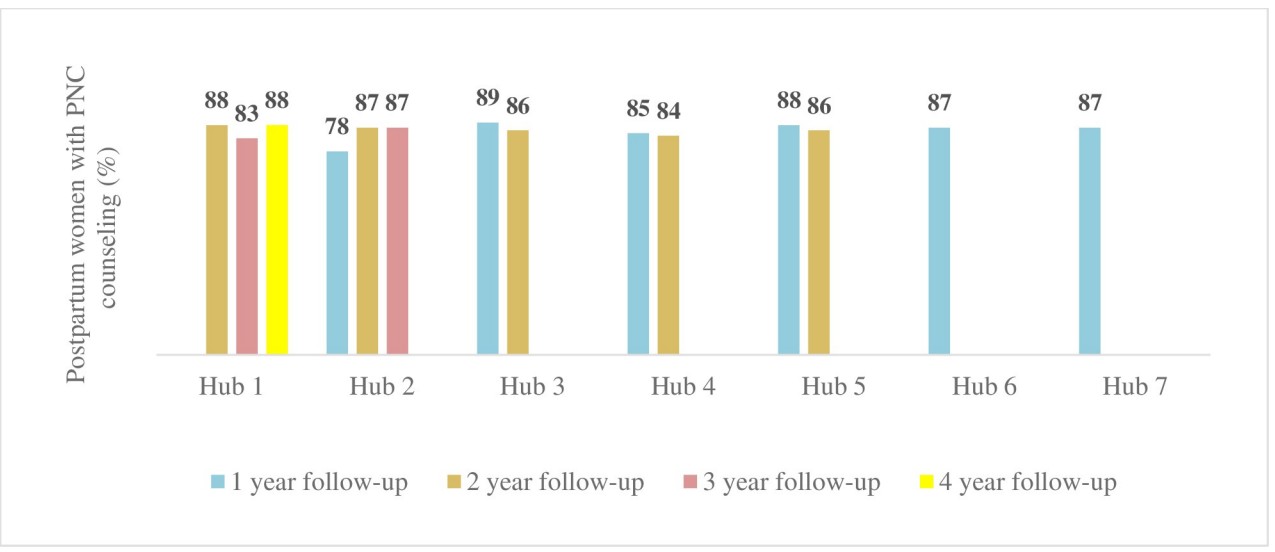

**Fig 4. Postnatal counseled home visits within 60 days of birth.**

population in Dolakha and 34% in Achham consist of *dalit*, *janjati*, or other castes [29]. The study data showed an enrollment of 50% families identifying as *dalit*, *janjati*, or other castes in Dolakha and 27% in Achham, suggesting high reach to marginalized families.

Our qualitative findings showed the CHW intervention reached hard-to-reach communities in our study catchment areas: *"It takes so much time to reach there, ma'am. The way is dangerous. [The way] is through jungles. I also went there once on the field visit. I was so scared. It was so dangerous walking through jungles, hills, and rivers, but sisters [CHWs] have not complained"* (CHN#1).

We also looked at the number of home visits in which the CHWs were able to provide one-year postpartum women PNC counseling within a week and 60 days of birth. About 78–89% of postpartum women received at least one CHW counseled postnatal visit within 60 days of birth (Fig 4), and 23–60% received postnatal counseling within a week of childbirth post-intervention (Fig 5).

**Effectiveness.** As a secondary effectiveness outcome, we assessed four or more facility ANC visit completion rates pre- and post-intervention. However, post-intervention facility ANC completion data were incomplete due to technical issues in the mobile application early in the study (S1 Fig). We later added a separate question asked after birth to collect self-reported facility ANC visit completion data. Wilcoxon signed-rank test of non-missing self-reported data suggested an average of 6% absolute change in facility ANC completion rates post-intervention across five hubs with available ANC data (p <0.001).

Our qualitative data also suggested a perceived increase in facility ANC visits with intervention implementation. One CHN stated: *"One of the things I have noticed in these two years is that, first of all, there has been a lot of improvement at the community level. The ANC visit rate has increased a lot. ANC visit rate was low in the past; now, they would go every month"* (CHN #3).

**Adoption.** In the original study plan, intervention implementation was planned for 19 municipalities across Achham and Dolakha. Of these, 10 (53%) municipalities adopted the program, and 7 (37%) municipalities adopted it within the intended timeline. Ramaroshan municipality in Achham that did not adopt the intervention within the intended timeline was later interested in replicating the program with some changes and technical support from

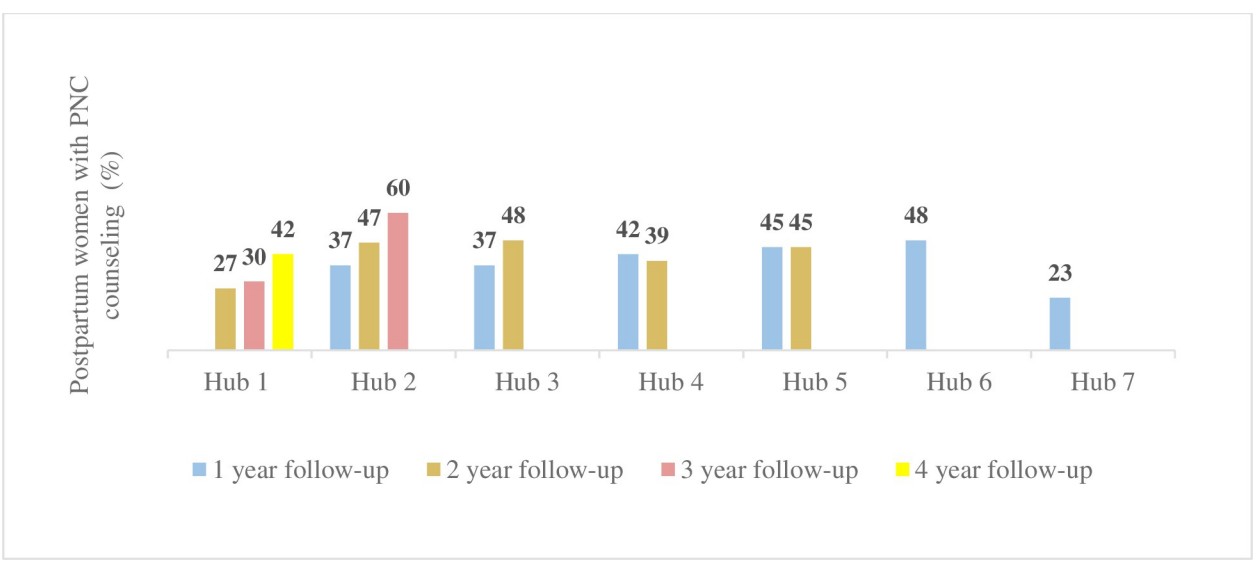

**Fig 5. Postnatal counseled home visits within a week of birth.**

NHN. The local government in Ramaroshan supports CHWs through municipal resources unlike other study catchment areas. One program staff noted:

> *"[The Vice-chairperson from Ramaroshan municipality] showed a lot of interest in the program. She said the program was very good. They were ready to work together and asked about costs and what needed to be done. We started planning after that. There was a discussion that the CHW program would belong to the Rural Municipality, and we would provide technical support only" (Program staff #1).*

**Implementation.** We evaluated the fidelity of the home-based ANC intervention. Per the research protocol, each pregnant woman would receive 6–8 home visits per pregnancy, from pregnancy diagnosis to birth. If women approached for a home visit were unavailable for counseling, the CHW would return in two weeks for another attempt. We looked at the number of home visits in which the CHWs were able to provide counseling. Our findings showed that in the first year of implementation across all hubs, the median counseled ANC home visits per pregnancy ranged from 3 to 5. In all hubs with additional follow-up data, the median improved to a range of 5 to 8 counseled home visits with additional implementation time (Table 3).

We measured the total immediate referrals made to health facilities for high-risk conditions during the antenatal and postnatal (within 60 days postpartum) periods. Reasons for immediate ANC referrals included vaginal bleeding, vaginal fluid leakage, abdominal pain, high or low blood pressures, post estimated delivery date, and anemia. Reasons for immediate PNC referrals included heavy vaginal bleeding, shortness of breath or chest pain, concerns for infection in the uterus, urinary tract, or breast, and high or low blood pressures. If the same women were referred multiple times for the same or different high-risk conditions during the same pregnancy, they were counted separately towards the total referrals. There were stark differences in the total number of immediate referrals between these hubs due to differences in hub sizes. Hub 1, the largest hub with 14 wards, had the highest referrals. There seemed to be

**Table 3. Median and interquartile range of counseled antenatal care home visits.**

| Hubs | 1-year follow-up | 2-year follow-up | 3-year follow-up | 4-year follow-up |
|------|------------------|------------------|------------------|------------------|
| Hub 1 | -- | 5 [4, 6] | 5 [3, 6] | 5 [4, 7] |
| Hub 2 | 5 [3, 7] | 8 [6, 9] | 7 [6, 9] | -- |
| Hub 3 | 4 [2, 5] | 6 [4, 7] | -- | -- |
| Hub 4 | 4 [2, 6] | 5 [3, 7] | -- | -- |
| Hub 5 | 5 [3, 6] | 6 [5, 7] | -- | -- |
| Hub 6 | 4 [2, 6] | -- | -- | -- |
| Hub 7 | 3 [2, 5] | -- | -- | -- |

higher PNC referrals than ANC referrals in the first year in most hubs. For hubs with more than one year of data, PNC referrals seemed to decrease in the later years (Table 4).

We qualitatively explored implementation barriers and facilitators with the implementing team. Several participants cited the most common challenge in providing timely care as the rainy season, which would sometimes exacerbate the adverse road conditions and cause power outages. As one CHPA stated: *"During the rainy season, they [CHWs] call saying that it is difficult to go to the field. Sometimes, the cell phone battery dies as there are a lot of electricity problems here during the rainy season"* (CHPA #1).

As noted by several implementing team members, a lack of trust during enrollment also posed some initial challenges in the implementation. However, they expressed the longitudinal nature of the intervention helped the CHWs build trust over time and encouraged women to open up and seek services: *"In the beginning, they didn't trust us, but as the CHWs went monthly to their home, their trust built up. Even if they couldn't share things at home at first, they can call CHWs and share them. As soon as something happens, they would call CHWs and ask what they should do"* (CHN #3).

Participants expressed the mobile tool was a facilitator in providing care despite some occasional technical issues. As one CHW noted: *"It has made things easier. Until it runs smoothly, it*

**Table 4. Total antenatal and postnatal immediate referrals made for high risk conditions.**

| Hubs | 1-year follow-up | | | 2-year follow-up | | | 3-year follow-up | | | 4-year follow-up | | |
|------|------------------|---|---|------------------|---|---|------------------|---|---|------------------|---|---|
| | Total births (n) | ANC referrals counseling (n) | PNC referrals counseling (n) | Total births (n) | ANC referrals counseling (n) | PNC referrals counseling (n) | Total births (n) | ANC referrals counseling (n) | PNC referrals counseling (n) | Total births (n) | ANC referrals counseling (n) | PNC referrals counseling (n) |
| Hub 1 | -- | -- | -- | 661 | 164 | 60 | 571 | 117 | 45 | 551 | 62 | 40 |
| Hub 2 | 552 | 27 | 48 | 450 | 62 | 51 | 404 | 60 | 23 | -- | -- | -- |
| Hub 3 | 265 | 15 | 6 | 300 | 15 | 9 | -- | -- | -- | -- | -- | -- |
| Hub 4 | 557 | 67 | 89 | 458 | 48 | 20 | -- | -- | -- | -- | -- | -- |
| Hub 5 | 137 | 20 | 29 | 124 | 34 | 12 | -- | -- | -- | -- | -- | -- |
| Hub 6 | 543 | 71 | 83 | -- | -- | -- | -- | -- | -- | -- | -- | -- |
| Hub 7 | 247 | 40 | 43 | -- | -- | -- | -- | -- | -- | -- | -- | -- |
| **Total** | **2301** | **240** | **298** | **1993** | **323** | **152** | **975** | **177** | **68** | **551** | **62** | **40** |

*facilitates our work and there are problems when it doesn't work well. Problems are not that frequent. We can see whom to meet and when to meet. It has been guiding us" (CHW FGD #3).*

The implementation process was iterative and adaptive to the evolving programmatic needs over time. Details of these adaptations in the intervention will be presented in a forthcoming manuscript. The iterative implementation process also led to multiple changes in forms, requiring adaptations of data quality checks, data reviews, and feedback loops.

**Maintenance.** To assess the sustainability of the intervention, we conducted a retrospective costing analysis for the first two hubs. The findings showed that the population-weighted average annual programmatic cost per capita of the ANC and PNC components only was US $1.19 across the hubs, and US $2.53 for the overall integrated RMNCH intervention (published in a separate manuscript) [14].

The integrated intervention has been maintained for more than one year beyond the study period (April 2020 to present), despite the COVID-19 pandemic. It was also expanded to four additional hubs after the study period and thus is currently operational in 11 hubs across both districts. However, there have been some adjustments in the CHW supervision structure and management after the study period. In Achham, one CHPA used to oversee 2–3 CHNs, whereas currently, one CHPA supervises eight CHNs. In Dolakha, the management of the CHW program has transitioned to a different not-for-profit hospital. With this change in management, two CHPA and six CHN positions were eliminated, seven CHWs resigned, and six new CHWs were hired.

In addition, the Nursing and Social Security Division under the Nepal Ministry of Health and Population is also poised to implement a similar community based integrated RMNCH intervention using staff nurses in two other districts in different regions of Nepal.

## Mechanisms of impact of the intervention

Participants perceived CHW counseling contributed to enhanced knowledge that led to positive health-seeking behavior and health outcomes among women. Participants also perceived high-risk pregnancy identification and timely referrals helped decrease morbidity and mortality by escalating care for pregnancy complications. Some program participants perceived women benefited from group counseling because it fostered social support and a safe space to share their problems (Table 5). While some CHWs shared encounters where women declined to receive multiple counseling sessions, most felt longitudinal and repeated counseling provided by the CHWs was effective because it helped women understand and retain information better. (Table 5).

## Discussion

We evaluated the effectiveness and the implementation of the integrated RMNCH intervention delivered by full-time, salaried, trained, and supervised CHWs in rural Nepal. This study showed an average 30% increase in IBR (p <0.0001), after adjusting for confounding variables. Due to our data limitations, we did not run adjusted model for facility ANC completion rates, but Wilcoxon signed rank test showed 6% absolute change (p <0.001) post-intervention. Future studies should attempt to understand more about the impact on facility ANC completion rates. Additionally, our data show high reach in marginalized populations, moderate implementation fidelity and adoption, and successful maintenance and expansion after the study period. Our findings suggest that the integrated intervention delivered by CHWs may be an effective approach in improving maternal healthcare utilization and outcomes in underserved rural settings. The perceived mechanisms by which the intervention impacted the outcomes were increased participant knowledge, timely referrals for complications, peer support,

**Table 5. Illustrative quotes of perceived mechanisms of impact of the intervention.**

| | |
|---|---|
| Increased knowledge | "*When we look at today's situation and, in the past, there are a lot of differences. Like who was pregnant, we would only know after 4–5 months. We could see her stomach out and know that she was pregnant. Today, if women don't get their periods within three days of their date, they come to us for a urine test*" (CHW FGD #1).<br>"*They didn't know much about what iron was. They also didn't know much about caring for the baby; the umbilical cord would be infected; there would be sores in the body and they would massage with oil in that condition. They had no concept about delivering the baby at a health facility. A lot has changed in how they (pregnant women, new mothers and children) receive care*" (CHW FGD #3). |
| Timely referrals for high-risk condition | "*We do early detection. For example, we have ultrasounds in eighth and ninth months. We directly refer for any complications, and the CHWs visit that ANC mother every month. If there are any problems, they [CHWs] immediately send them [pregnant women] to the hospital. Because of that, the mortality and morbidity of ANC have reduced and that was shared by the rural municipality*" (CHN #2). |
| Peer Support | "*They are getting counseling in a group. They are getting services from their own sisters [CHWs]; it was really nice to do problem sharing at that time. They would become happy*" (CHPA #1). |
| Longitudinal counseling | "*Of course, I remember such things, but the more they share, the more we will remember*" (Reproductive-aged women #4).<br>"*We tell them to sit for a while and listen to something and go; they say they have got some work to do. Some people say how much they should listen every month*" (CHW FGD #1). |

longitudinal CHW counseling, use of a mobile tool for CHWs, and trust between CHWs and community members.

Our findings were consistent with another quasi-experimental study of community-based ANC interventions delivered by CHWs in rural Tanzania, which showed increased institutional delivery (76% to 90%) two years after the intervention [30]. Other studies in similar settings also showed CHWs providing care using digital tools and linking them with health facilities contributed to improved institutional delivery and uptake of ANC services [31–33].

Our findings suggest high reach to marginalized families; however, since our analysis was at the ward level, we were unable to compare care utilization of individuals by caste. Further study is warranted to understand any disparities in utilization and health outcomes. Other CHW studies in similar settings have shown disparities in service utilization and health outcomes despite high CHW reach to marginalized populations [34–36].

Our findings showed CHW postnatal home visits reach within 60 days of birth and a week of birth were 78–89% and 23–60%, respectively. While the WHO and the GoN recommend postnatal visits within a week of birth, a workflow for the CHWs completing this visit was not built into the original protocol for the integrated intervention. Given that the national indicators showed low PNC visit completion rates within a week of childbirth [37], the implementation team later recognized that the CHWs could augment government services by making a home visit within a week of birth. Therefore, an effort was made to do so towards the final year of the study. This change in implementation was challenging due to high CHW workloads in addition to the potential seasonal barriers in providing timely care. This area warrants further exploration within a CHW model.

Nepal's evolving political context, namely the federal system and decentralization in 2017, affected intervention adoption by municipalities. Decentralization resulted in significant changes in the decision-making landscape [38], whereby decision-making by the central government prior to the study onset shifted to local governments during the stepped implementation. Additionally, the delayed timelines in implementation were partly due to the extra time it took to engage with newly elected local stakeholders. While the shift in decision-making likely

led to the decreased adoption rate and delays in implementation, it also resulted in opportunities to create new local ownership, exemplified by the most recent initiative of Ramaroshan municipality in Achham to replicate the integrated CHW intervention.

With regard to fidelity of the intervention, our findings suggest the median number of ANC home visits where counseling took place improved over time in the hubs where we had follow-up data beyond one year. This may have been because CHWs learned the best time to reach women at home, and women knew when to expect CHWs and became more accepting of their counseling over time. Our qualitative data suggested that trust between CHWs and women developed gradually over time with longitudinal interactions. Furthermore, in the first year of follow-up data across hubs, the number of PNC referrals seemed higher than the ANC referrals, driven largely by referrals for shortness of breath, chest pain, or dizziness. These PNC referrals in the later years of follow-up data seemed to decrease, perhaps due to improvements in accurate recognition of symptoms by CHWs with more experience.

Given the presence of the government FCHV cadre throughout the study catchment areas, the implementing team attempted to have CHWs work closely with the FCHVs to coordinate care. CHWs participated in monthly FCHV meetings to compare data and ensure better reach. FCHVs would sometimes request CHWs to follow up on pregnancy tests or refer children to immunization centers. We recommend further study to understand and strengthen collaboration between FCHVs and CHWs.

The effectiveness and the cost make this cadre of CHWs a sustainable workforce beyond the study period, and thus the program is currently operational in 11 hubs. However, some adjustments to the CHW supervision, such as reducing the number of CHPAs and CHNs, while strategic for cost-saving, likely also have potential implications on work satisfaction, supervision, and retention. There may also be an impact on the implementation of the integrated intervention and health outcomes. Further study is warranted to understand the impact of alternative management and supervision structures for CHWs.

In rural Nepal settings, structural and socio-cultural factors, including poverty, healthcare delivery gaps, gender inequality, low literacy rates, and challenging terrain, play roles in maternal health outcomes [37,39,40]. While our study addressed health literacy, community-based care, social support for women, and linkages between community and facility-based care, other structural and socio-cultural factors, such as facility quality and gender-based violence that also affect health outcomes were beyond the scope of this study. In the long-term, improvements in the healthcare quality coupled with other community efforts to address social determinants of health may also be essential for substantial impact [41–43]. The findings from this study could potentially inform community health strategy and scale-up in Nepal and similar settings.

## Limitations

There were several limitations to this study. We chose a quasi-experimental trial design as an ethical, pragmatic, acceptable, and affordable approach in our context in rural Nepal. Without a randomized controlled design, we were unable to establish the causality between the CHW intervention and observed outcomes. We did not reach the anticipated 300,000 catchment population within the expected study timeline largely due to Nepal's shift to federalism during the study period. Furthermore, our quantitative data collection was cut a few months short due to the COVID-19 pandemic and resulting lockdowns. The reduced sample size and data collection period likely had implications on our statistical power.

The initial lack of trust by the community members during enrollment might have caused lower reporting of events at baseline, although the CHWs were able to gradually

build trusting relationships by providing longitudinal care. The use of birth histories, asking women to recall past events for baseline data, might have also led to recall bias. Of note, to investigate this limitation further, we conducted a comparison of birth history data with programmatic data on births. We found that although more births were identified in programmatic data, reporting on birth location was largely consistent among births identified by both methods [25].

The use of programmatic data collected by CHWs, who were employed to improve community health outcomes, might have potentially introduced bias in data reporting [44–46]. However, routine monitoring and supervision, data quality checks, data review and verification throughout the study period likely mitigated these effects in our study.

The iterative implementation process led to multiple changes in CommCare forms and workflows, resulting in missing or incomplete data. Missing data posed challenges in data processing and analysis of some originally planned variables, and some available data were not adequate to derive conclusions. We excluded unavailable or missing data for implementation variables, which might have introduced bias in the results. Our facility antenatal completion data might have been especially affected, which was why we could not perform rigorous analysis beyond a simple non-parametric analysis.

## Conclusions

Our findings show that community-based RMNCH services delivered by full-time, paid, supervised, and trained CHW cadre may be effective in improving maternal health in rural low- and middle-income settings rife with health system and workforce challenges. Therefore, settings grappling with similar challenges could utilize CHWs systems to reduce health disparities and improve universal health coverage in maternal and reproductive health. The implementation outcomes from our study could help plan future implementation processes in similar settings effectively. Given that both study districts are rural and have challenging terrains, the findings may not be generalizable to urban and high-resource contexts. However, for many other LMICs that face similar challenges, and other communities in Nepal, this study can offer insights into designing and implementing CHW systems for improving reproductive and maternal health.

## Supporting information

**S1 Fig. Self-reported facility antenatal care completion rates.**
(TIF)

**S1 Text. Inclusivity in global research.**
(DOCX)

## Acknowledgments

We acknowledge our appreciation to the Nepal Ministry of Health and Population for their continued efforts to improve public healthcare system in Nepal including their contribution in setting this study up for success. We would also like to acknowledge inputs from Dr. Pushpa Chaudhary, former Health Secretary in Ministry of Health and Population, Nepal, in designing and improving the intervention. We are deeply grateful to and inspired by the community and hospital staff, including community health workers, nurses, whose dedication, and service continue to create positive health impact in rural Nepal.

## Author Contributions

**Conceptualization:** Duncan Maru, Isha Nirola, Sheela Maru.

**Data curation:** Aparna Tiwari, Nandini Choudhury, Laxman Datt Bhatt, Ved Bhandari, Kshitiz Rana Magar.

**Formal analysis:** Aparna Tiwari, Nandini Choudhury, Rekha Khatri, Samrachana Adhikari.

**Funding acquisition:** Duncan Maru.

**Investigation:** Aparna Tiwari, Aradhana Thapa, Nandini Choudhury, Rekha Khatri, Sabitri Sapkota, Wan-Ju Wu, Sheela Maru.

**Methodology:** Wan-Ju Wu, Scott Halliday, David Citrin, Duncan Maru, Isha Nirola, Poshan Thapa, Sheela Maru.

**Project administration:** Aparna Tiwari, Aradhana Thapa, Nandini Choudhury, Rekha Khatri, Sabitri Sapkota, Wan-Ju Wu, Scott Halliday, David Citrin, Ryan Schwarz, Duncan Maru, Hari Jung Rayamazi, Rashmi Paudel, Laxman Datt Bhatt, Ved Bhandari, Nutan Marasini, Sonu Khadka, Bhawana Bogati, Sita Saud, Yashoda Kumari Bhat Kshetri, Aasha Bhatta, Kshitiz Rana Magar, Ramesh Shrestha, Ranjana Kafle, Roshan Poudel, Samiksha Gautam, Indira Basnett, Goma Niroula Shrestha, Isha Nirola, Poshan Thapa, Lal Kunwar, Sheela Maru.

**Software:** Nutan Marasini.

**Visualization:** Aparna Tiwari, Nandini Choudhury.

**Writing – original draft:** Aparna Tiwari.

**Writing – review & editing:** Aradhana Thapa, Nandini Choudhury, Rekha Khatri, Sabitri Sapkota, Wan-Ju Wu, Scott Halliday, David Citrin, Ryan Schwarz, Duncan Maru, Hari Jung Rayamazi, Rashmi Paudel, Laxman Datt Bhatt, Ved Bhandari, Nutan Marasini, Sonu Khadka, Bhawana Bogati, Sita Saud, Yashoda Kumari Bhat Kshetri, Aasha Bhatta, Kshitiz Rana Magar, Ramesh Shrestha, Ranjana Kafle, Roshan Poudel, Samiksha Gautam, Indira Basnett, Goma Niroula Shrestha, Isha Nirola, Samrachana Adhikari, Poshan Thapa, Lal Kunwar, Sheela Maru.

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
