## [Decision Letter · Decision Letter 0]

23 Aug 2022

PGPH-D-22-00519

A Type II hybrid effectiveness-implementation study of an integrated CHW intervention to address maternal healthcare in rural Nepal

Dear Dr. Tiwari,

Thank you for submitting your manuscript to PLOS Global Public Health. After careful consideration, we feel that it has merit but does not fully meet PLOS Global Public Health’s publication criteria as it currently stands. Therefore, we invite you to submit a revised version of the manuscript that addresses the points raised during the review process.

We look forward to receiving your revised manuscript.

Kind regards,

Siyan Yi, MD, MHSc, PhD

Academic Editor

Journal Requirements:

2. Please amend your online detailed Financial Disclosure statement. This is published with the article. It must therefore be completed in full sentences and contain the exact wording you wish to be published.

Please state what role the funders took in the study. If the funders had no role in your study, please state: “The funders had no role in study design, data collection and analysis, decision to publish, or preparation of the manuscript.”

3. Please update the completed 'Competing Interests' statement. Please declare all competing interests beginning with the statement “I have read the journal's policy and the authors of this manuscript have the following competing interests:”.

4. In the online submission form, you indicated that your data will be submitted to a repository upon acceptance.  We strongly recommend all authors deposit their data before acceptance, as the process can be lengthy and hold up publication timelines. Please note that, though access restrictions are acceptable now, your entire data will need to be made freely accessible if your manuscript is accepted for publication. This policy applies to all data except where public deposition would breach compliance with the protocol approved by your research ethics board. If you are unable to adhere to our open data policy, please kindly revise your statement to explain your reasoning and we will seek the editor's input on an exemption. Please be assured that, once you have provided your new statement, the assessment of your exemption will not hold up the peer review process.

Additional Editor Comments (if provided):

Reviewers' comments:

Reviewer's Responses to Questions

**Comments to the Author**

1. Does this manuscript meet PLOS Global Public Health’s publication criteria? Is the manuscript technically sound, and do the data support the conclusions? The manuscript must describe methodologically and ethically rigorous research with conclusions that are appropriately drawn based on the data presented.

Reviewer #1: Partly

Reviewer #2: Yes

2. Has the statistical analysis been performed appropriately and rigorously?

Reviewer #1: Yes

Reviewer #2: Yes

3. Have the authors made all data underlying the findings in their manuscript fully available (please refer to the Data Availability Statement at the start of the manuscript PDF file)?

Reviewer #1: No

Reviewer #2: No

4. Is the manuscript presented in an intelligible fashion and written in standard English?

Reviewer #1: Yes

Reviewer #2: Yes

5. Review Comments to the Author

Reviewer #1: This study assesses the effectiveness and the implementation of an evidence-based integrated maternal and child health intervention delivered by CHWs, using a mixed method. Data and statistical methods used are not clearly described. The document does not present clearly the characteristics of participants, which could also be associated with the outcome variables. It is important to consider seasonality of data collected because fertility and child mortality could vary by month.

Reviewer #2: Overall, this is an interesting paper that shows excellent use of implementation science in program implementation. I have few comments:

Abstract

Sentence on what it means for wider public health or contribution to general bidy of knowledge seems to be missing.

Introduction

Line 108 - What ‘contextual challenge’ is being referred to please be more specific.

Line 112/113 – please add citation.

Line 131 – what were the retention challenges? Also add citation.

Line 135 – Did the project look at specific components of evidence-based intervention? If so, please add.

Line 141 – reach number seems to be high. Is this for maternal health? Based on Census and HMIS data, I don’t think total size of two districts will add up to 300,00. Please explain, if there is double counting and what events are counted.

Study Population

Line 177 – What was the percentage of excluded population

Line 179 - How were the three groups recruited for the study. Please add sampling process. Also who were included in the program staff, is it just CHPA?

Intervention

Line 204 - it the sentence referring to ‘post-partum contraception’, if so specify this.

Line 205 – Please describe what components were included in the mobile tools, was it counselling and referral only or were there other functions as well

Line 213 – Some of the references in the intervention section seem to be from earlier published articles of the program. Is there a way to clarify this.

Study design and Implementation approach

Line 230 – Add some description of RE-AIM

It would be good to have table that shows key question for each phase of RE-AIM to help reader understand overall objective. This can include process and target respondent for each phase

Line 235 – How were the clusters formed? Were there any specific criteria since the number of wards in cluster seems to vary

Line 243 – was the enrollment period in each hub continuous through the life of project. Line 234 seem to imply it was not. If so, how were the new pregnancies beyond the enrollment period in a hub managed? What about post PNC period, how were they managed? Were there any follow-up post that period?

Line 248 – what was the overall attrition rate, did it impact the study in any way?

Data Analysis

Line 313 – Besides self-reporting, were there any CHWs record/reports that were used throughout the process?

Study Enrollment

Enrollment of Dalit/Janajati/others in Hub 2 and 7 seems to be higher compared to other hubs. Is there a reason for this?

Line 357/358 – What was the reason for not being able to process data from three hubs?

Implementation Outcome

Line 397 – It seems women were counted more than once. Please explain.

Line 463 – What percentage were counted separately? Did this have any impact on the study.

Line 480 – ‘lack of trust’ please mention this in enrollment or in limitation

Mechanism of impact of the intervention

Were there any differences in findings by type of respondents – if so it would be good to specify this.

Discussion

Line 545 – ‘high reach’ – Was this not by design, earlier table seem to suggest this.

Line 551 – What was the reason for wide range in week of birth?

Discussion section seems to have more details on findings. Some section is missing comparison from other published articles or findings from similar contexts.

Also add what this means for wider public health context. How should this body of knowledge be treated.

6. PLOS authors have the option to publish the peer review history of their article (what does this mean?). If published, this will include your full peer review and any attached files.

**Do you want your identity to be public for this peer review?** For information about this choice, including consent withdrawal, please see our Privacy Policy.

Reviewer #1: **Yes: **Jacques Emina Be-Ofuriyua

Reviewer #2: No

---

## [Editor Report · Decision Letter 1]

28 Dec 2022

A Type II hybrid effectiveness-implementation study of an integrated CHW intervention to address maternal healthcare in rural Nepal

PGPH-D-22-00519R1

Dear Ms. Tiwari,

We are pleased to inform you that your manuscript 'A Type II hybrid effectiveness-implementation study of an integrated CHW intervention to address maternal healthcare in rural Nepal' has been provisionally accepted for publication in PLOS Global Public Health.

Best regards,

Siyan Yi, MD, MHSc, PhD

Academic Editor